# Potential of Intestinal Current Measurement for Personalized Treatment of Patients with Cystic Fibrosis

**DOI:** 10.3390/jpm11050384

**Published:** 2021-05-08

**Authors:** Simon Y. Graeber, Constanze Vitzthum, Marcus A. Mall

**Affiliations:** 1Charité—Universitätsmedizin Berlin, Department of Pediatric Respiratory Medicine, Immunology and Critical Care Medicine, 13353 Berlin, Germany; simon.graeber@charite.de (S.Y.G.); constanze.vitzthum@charite.de (C.V.); 2Berlin Institute of Health, Charité—Universitätsmedizin Berlin, 10117 Berlin, Germany; 3German Centre for Lung Research (DZL), 13353 Berlin, Germany

**Keywords:** cystic fibrosis, CFTR, intestinal current measurement (ICM), personalized medicine

## Abstract

Refinement of personalized treatment of cystic fibrosis (CF) with emerging medicines targeting the CF basic defect will likely benefit from biomarkers sensitive to detect improvement of cystic fibrosis transmembrane conductance regulator (CFTR) function in individual patients. Intestinal current measurement (ICM) is a technique that enables quantitative assessment of CFTR chloride channel function in rectal tissues or other intestinal epithelia. ICM was originally developed to study the CF ion transport defect in the intestine and has been established as a sensitive biomarker of CFTR function and diagnostic test for CF. With the emergence of CFTR-directed therapeutics, ICM has become an important tool to estimate the level of rescue of CFTR function achieved by approved CFTR modulators, both at the level of *CFTR* genotype groups, as well as individual patients with CF. In combination with preclinical patient-derived cell culture models, ICM may aid the development of targeted therapies for patients with rare *CFTR* mutations. Here, we review the principles of ICM and examine how this CFTR biomarker may be used to support diagnostic testing and enhance personalized medicine for individual patients with common as well as rare *CFTR* mutations in the new era of medicines targeting the underlying cause of CF.

## 1. Introduction

Recent breakthroughs in cystic fibrosis (CF) transmembrane conductance regulator (CFTR)-directed therapies have heralded a new era of precision medicine for patients with CF [1,2,3,4,5,6,7,8,9,10,11,12,13]. Biomarkers that are sensitive to detect improvement of CFTR function in individual patients could help to assess individual treatment responses to these therapies targeting the underlying CF defect. In addition to sweat chloride concentration as a measurement of CFTR function in the sweat duct, intestinal current measurement (ICM) has been developed as a technique that enables quantitative assessment of CFTR chloride channel function in native intestinal tissues. Shortly after the discovery of the *CFTR* gene in 1989 [14], several research groups developed modified Ussing chambers for transepithelial measurements across native tissues from the rectum and other regions of the intestinal mucosa to study the CF ion transport defect in the intestine [15,16,17,18,19,20,21,22,23,24,25,26,27,28,29,30,31]. At this time, the focus was on understanding the role of CFTR as a cAMP-dependent chloride channel in ion and fluid transport in the gastro-intestinal tract, and the effect of CFTR dysfunction caused by mutations in the *CFTR* gene on this process. In these studies it was found that, in contrast to the airways where CFTR is expressed together with the alternative calcium-activated chloride channel TMEM16A that remains intact in CF [32,33,34,35,36,37,38,39], CFTR is the dominant chloride channel responsible for chloride and fluid secretion in human colon [18,21,22,24,27,30]. The characterization of CFTR dysfunction in these studies led to the development of experimental protocols that were able to differentiate between impaired CFTR function in intestinal tissues from patients with CF vs. normal CFTR function in control subjects [29,31]. In addition, protocols were developed that enabled the classification of CF tissues into two groups with (i) lack of or minimal detectable CFTR function and (ii) residual CFTR function [28,29,31,40]. Based on these results, ICM was established as a diagnostic test to aid in establishing or refuting a diagnosis of CF, if sweat test results are equivocal and/or if the functional consequence of rare or newly detected *CFTR* variants are unknown. With the emergence of CFTR-directed therapeutics in the clinical arena, sensitive biomarkers of CFTR function such as ICM have the potential to facilitate and enhance personalized therapy for patients with CF [1,2,41,42]. In this context, ICM was shown to be sensitive to detect improvement of CFTR function when intestinal tissue biopsies were treated with CFTR modulators ex vivo or obtained from patients with CF that were treated systemically with CFTR modulators [43,44,45,46]. In this review, we describe the principles of ICM, summarize its use as a sensitive biomarker of CFTR function, and examine recent studies using ICM as an outcome measure of response to CFTR-directed therapeutics, and how the use of ICM may contribute to further improvement in personalized medicine eventually for all patients with CF.

## 2. Principle of Intestinal Current Measurement (ICM)

For ICM, small superficial biopsies of the rectal mucosa (i.e., only epithelial layer) are obtained by forceps or suction biopsy [21,29,31]. Collection of the biopsies is painless, does not require sedation and is safe and easy to perform at any age. Rectal tissues are immediately mounted in perfused or circulating small aperture Ussing chambers and equilibrated in physiological buffer solution [21,29,31]. After equilibration, basal bioelectric properties are recorded under open-circuit or short-circuit conditions, and the effects of pharmacological stimuli on intestinal ion transport can be studied ex vivo in a controlled setting. Initially, two different setups of micro-Ussing chambers were developed to measure transepithelial currents after pharmacological activation of CFTR in native rectal tissues. The setup originally developed at the University Hospital of Rotterdam, The Netherlands, uses traditional re-circulating micro-Ussing chambers with a pharmacological protocol focusing mainly on calcium-mediated activation of the tissues to induce chloride secretion via CFTR [18,19,26,31]. The other setup, originally developed at the University of Freiburg, Germany, uses continuously perfused micro-Ussing chambers that enable studying the effects of pharmacological modulation of CFTR function under different experimental conditions (e.g., in the absence and presence of endogenous CFTR activation) in the same tissue in a strictly paired fashion. Pharmacological protocols with this setup enable the sequential assessment of cAMP-mediated and calcium-mediated chloride secretion as a readout of CFTR function [21,24,27,29,30,47,48]. Typically, ICM is performed in the presence of amiloride to block epithelial sodium channel (ENaC)-mediated sodium absorption and indomethacin to inhibit cyclooxygenase activity and, thus, suppress the synthesis of prostaglandins that stimulates endogenous cAMP formation in intestinal tissues [22,24].

Compared to airway tissues, rectal biopsies can be easily obtained by minimally invasive procedures and rectal tissue has several advantages for studies of CFTR function. First, rectal tissues express higher levels of CFTR resulting in a high signal-to-noise ratio. Second, in contrast to the airways, the rectal epithelium does not express alternative calcium-activated chloride channels [21,27], and, therefore, both cAMP- and calcium-mediated chloride secretion are strictly related to CFTR function. Finally, the intestine including the rectum is not affected by chronic inflammation, infection with CF pathogens or structural organ damage and remodeling, i.e., factors that may impede CFTR chloride channel function independent of the basic molecular defect of *CFTR* mutations.

## 3. ICM as a Biomarker of CFTR Function and Diagnostic Test for CF

Based on the results of the early studies on the pathophysiology of the CF ion transport defect in rectal tissue, standardized ICM protocols for pharmacological stimulation of CFTR chloride channels were established. These protocols were shown to differentiate between normal CFTR function in control subjects and a spectrum of CFTR dysfunction in CF, ranging from the complete absence of CFTR-mediated chloride secretion to substantial residual CFTR function detected by ICM in patients with different *CFTR* genotypes [28,40,47,49]. Specifically, it was found that in rectal tissues from healthy individuals, cAMP-dependent activation induces a sustained CFTR-mediated chloride secretory response [21]. This chloride secretory response is augmented by cholinergic co-activation with carbachol via activation of calcium-dependent potassium channels in the basolateral membrane of colonocytes that leads to an increase in the driving force for CFTR-mediated chloride secretion across the luminal membrane [37]. In patients with CF, the responses detected by ICM allow to distinguish between two functional phenotypes. In the majority of patients (~85%), CFTR-mediated chloride secretory responses are absent and instead ICM shows an inverse bioelectric response following cAMP- and calcium-dependent stimulation that was found to reflect luminal potassium secretion [24]. In a series of studies, it was shown that CF patients with this ICM functional signature typically carry two ‘severe’ *CFTR* mutations (i.e., class I to III) [28,40,47,50]. Further, it was shown that this absence of CFTR function determined by ICM is associated with a classical CF phenotype including sweat chloride concentrations in the upper diagnostic range, progressive sino-pulmonary disease and exocrine pancreatic insufficiency [28,29,40,47,50]. In the remaining group of patients (~15%), ICM detects residual CFTR function, as evidenced by an attenuated chloride secretory response to cAMP-dependent stimulation, as well as cholinergic co-activation typically reflected by a biphasic response upon co-stimulation of rectal tissues with carbachol [28,47]. Studies on the relationship between CFTR function detected by ICM, *CFTR* genotype and CF phenotype demonstrated that patients in this group typically carry at least one ‘mild’ CFTR mutation (class IV and V) and that on average, residual CFTR function was associated with a milder form of CF characterized by sweat chloride concentrations in the lower diagnostic or intermediate range, long-term exocrine pancreatic sufficiency, later age at diagnosis, and less severe impairment in nutritional outcomes and lung function [28,51]. These studies support a prognostic value of ICM and led to the implementation of ICM in the diagnostic algorithm for CF, especially to aid establish or refute a diagnosis of CF in patients with equivocal sweat test or genetic testing results [52]. In this context, recent studies also demonstrated the potential usefulness of ICM in the diagnosis of atypical CF and CFTR related disease [53,54]. In the future, with the widespread implementation of CF newborn screening in many countries, ICM may also become a useful tool for the diagnostic workup of CF newborn screening positive infants with inconclusive diagnosis (CF-SPID) to determine disease liability and prognosis [52,55,56,57,58]. Collectively, these studies have established ICM as a sensitive biomarker of CFTR function in the intestinal epithelium. However, due to the requirement of specialized technical equipment and well-trained personnel, availability of ICM remains limited to reference centers.

## 4. ICM as Outcome Measure of In Vivo Response to CFTR-Directed Therapeutics

The development and recent approval of an increasing number of CFTR modulator drugs has created an unprecedented opportunity to treat the basic defect in a growing number of patients with CF [4,5,6,7,8,9,10,11,12,13]. However, clinical trials of CFTR modulators showed heterogeneous responses in clinical outcomes as well as sweat chloride concentration among patients with identical *CFTR* genotypes. These results suggest that additional sensitive biomarkers of CFTR function will be important to study the degree of functional rescue of *CFTR* mutations by different CFTR modulator drugs, both at the level of *CFTR* genotype groups as well as individual patients. We expect these studies will be instrumental to determine response to therapy at the level of the CF basic defect independent of disease severity and/or environmental factors that may confound complex clinical outcomes such as spirometry (FEV_1_ % predicted), and thereby enhance the potential of personalized therapy for CF.

In this context, initial ‘real world’ studies with approved CFTR modulators demonstrated that ICM is sensitive to detect in vivo activation of CFTR in individual CF patients with a *G551D* mutation treated with the CFTR potentiator ivacaftor [44]. Here, ICM detected activation of CFTR-mediated chloride secretion to a mean level of ~50% of normal in *G551D* CF patients after starting treatment with ivacaftor [44]. Further, ICM demonstrated that the combination of the CFTR corrector lumacaftor with the potentiator ivacaftor as the first CFTR modulator combination therapy approved for the treatment of CF patients homozygous for the common *F508del* mutation leads to partial rescue of CFTR activity to levels of ~15–20% of normal CFTR function in healthy controls [45,46]. Interestingly, improvement in CFTR function detected by ICM was also observed in patients that did not show improvement in FEV_1_ % predicted after initiation of lumacaftor-ivacaftor therapy [45]. In this study, ICM detected robust improvement of CFTR function in almost all patients and the variance of ICM was substantially lower than that of nasal potential difference (NPD; total chloride secretory response) and lung function measures [45]. These findings indicate that clinical outcome measures such as FEV_1_ % predicted that are commonly used as endpoints in clinical trials have limitations as outcome measure of response to therapy at the level of the CF basic defect, and highlight the importance of including sensitive biomarkers of CFTR function in an approach to personalized medicine of CF. As the number of approved CFTR modulator drugs containing different combinations of small molecule compounds continues to increase, even for patients with the same *CFTR* genotype, ICM may become a useful tool for quantitative comparison of efficacy of functional correction of *CFTR* mutations at the level of genotype groups, and potentially serve as a biomarker to aid optimization of personalized treatment of individual patients.

## 5. Potential Use of ICM for Personalized Medicine for Patients with Rare Mutations and High Unmet Need

Almost a decade ago, the potentiator ivacaftor was the first approved CFTR modulator for patients with CF with a *G551D* gating mutation [4]. It was shown that ivacaftor restores CFTR function to ~50% of wild-type levels and improved FEV1 % predicted by ~10% in patients with at least one *G551D* mutation [4,44]. In patients with the common *F508del* mutation or other processing (class II) mutations, ivacaftor alone failed to improve CFTR function or lung function. However, the combination of a corrector (lumacaftor or tezacaftor), which partially overcomes the folding defect of F508del CFTR with the potentiator ivacaftor led to an improvement in CFTR function and clinical outcomes in patients homozygous for the *F508del* mutation [5,6]. Recently, a triple combination of the two correctors elexacaftor and tezacaftor with the potentiator ivacaftor showed substantial clinical improvement in patients with CF with at least one *F508del* allele [10,11]. As a result, effective CFTR modulators have recently been approved and are becoming available for the treatment of approximately 90 % of *CFTR* genotypes [4,5,6,7,8,9,10,11,12,13]. Despite this unprecedented breakthrough in precision medicine of the underlying CF defect, there are still approximately 10% of patients with *CFTR* genotypes that cannot be treated with current CFTR-directed therapeutics. Therefore, preclinical development and testing of new therapies targeting rare *CFTR* mutations that do not respond to current CFTR modulators, or for which responsiveness remains unknown due to their low occurrence, remains important to further improve precision medicine and ultimately provide effective therapies for all patients with CF. In this context, ICM was shown to detect improvement in CFTR-mediated chloride secretion in response to stimulation with a CFTR potentiator in rectal tissue of patients with a broad range of residual CFTR function mutations (Y161C, V232D, R334W, T338I, I1234V, 3272-26 A > G, 3849 + 10 kb C > T, 4005 + 5727 A > G, G576A, F1052V, M1137R, 1898 + 3 A > G) [43]. These studies also showed that ex vivo testing of multiple compounds and compound combinations in freshly excised rectal tissue by ICM is limited by the number of rectal biopsies that can be obtained as well as their limited viability even under primary tissue culture conditions. However, in addition to ICM, rectal biopsies can also be used to generate patient-derived intestinal organoids as a versatile model system for preclinical testing of investigational compounds targeting rare *CFTR* mutations that are not eligible for treatment with an approved CFTR modulator [59,60]. In addition, nasal or bronchial epithelial cells collected by brushings can be expanded and differentiated under air-liquid interface conditions to enable testing of multiple drug candidates in primary airway epithelial cultures derived from individual patients harboring specific *CFTR* mutations [61,62,63,64,65,66,67,68,69,70]. Several preclinical studies in these patient-derived model systems showed that the triple combination of elexacaftor–tezacaftor–ivacaftor, as well as several novel modulator combinations improve CFTR function in class II mutations other than *F508del*, as well as other rare *CFTR* mutations [68,69,70,71,72,73,74]. Functional in vitro assays in these patient-derived intestinal and airway model systems were shown to correlate with biomarkers of CFTR function in patients with a spectrum of *CFTR* mutations associated with a broad range of CFTR modulator responses [62,75]. However, recent studies found no correlation between the magnitude of in vitro response in rectal organoids and in vivo response to CFTR modulator treatment detected by measurements of sweat chloride concentration, NPD or ICM in more homogenous patient groups receiving the same CFTR modulator [76,77]. These findings indicate that in vivo restoration of mutant CFTR function probably depends on multiple factors including pharmacokinetics that cannot be entirely predicted by these patient-derived in vitro models. Therefore, biomarkers of in vivo CFTR function remain important for the assessment of response to therapy also in individual patients, which can be achieved once the investigational compound can be administered to patients. Among the different in vivo biomarkers of CFTR function, sweat chloride concentration is the most broadly available and was shown to detect improvement even in *F508del* homozygous patients treated with lumacaftor-ivacaftor with moderate improvement of CFTR function [45,78,79]. However, in about 20% of *F508del* homozygous patients, improvement in sweat chloride concentration under treatment with lumacaftor-ivacaftor did not correspond with improvement of CFTR function detected by ICM or NPD [45]. These results suggest that ICM and NPD can capture additional, potentially organ-specific aspects of in vivo rescue of mutant CFTR function by CFTR modulator therapy. On note, in studies comparing NPD and ICM as in vivo biomarkers of CFTR function, ICM was superior to NPD in distinguishing between patients with CF and healthy controls, and ICM demonstrated substantially greater power than NPD to detect low levels of residual CFTR function [50,80]. In sample size calculations based on this study, as little as five patients may be sufficient to detect a ten percent improvement of CFTR function by ICM, highlighting the high sensitivity of ICM as well as its potential in early phase clinical trials [50]. Along these lines, in our previous real world study evaluating the effects of lumacaftor-ivacaftor in *F508del* homozygous patients with CF, we found that ICM detected improvement in CFTR function in 95% of patients compared to 65% of patients showing functional improvement in response to therapy by NPD [45]. Taken together, these results support ICM as a sensitive outcome measure for testing of in vivo efficacy of CFTR modulators. In combination with patient-derived in vitro cell culture models for preclinical testing of investigational compounds ICM may therefore facilitate the development of effective therapies of the basic defect for all patients with CF (Figure 1).

## 6. Conclusions

In summary, ICM is a valuable biomarker to characterize CFTR function in native intestinal tissues and a sensitive outcome measure of in vivo response to CFTR modulator therapy in patients with CF (Table 1). In patients with access to CFTR modulator therapies, ICM can therefore be used to determine the level of functional rescue, both at the level of genotype groups as well as individual patients. As an increasing number of CFTR modulators is becoming available for patients with common *CFTR* mutations, ICM may help to select the most efficacious available CFTR modulator for the individual patient. For patients with rare *CFTR* mutations with unknown functional consequences and no access to CFTR modulator treatment, functional characterization with ICM may help to identify potential treatment options. Further, rectal tissue collected for ICM can also be used for the generation of intestinal organoids for preclinical in vitro testing of novel drug candidates. Promising candidates may then be tested using ICM as sensitive CFTR biomarker to determine in vivo efficacy in early phase proof-of-concept clinical trials or n-of-1 studies especially for patients with rare *CFTR* mutations. We conclude that ICM, as a sensitive CFTR biomarker for quantitative assessment of response to CFTR-directed therapeutics at the level of the CF basic defect, has a high potential to facilitate enhancement of personalized medicine for patients with common as well as rare *CFTR* genotypes.

## Figures and Tables

**Figure 1 jpm-11-00384-f001:**
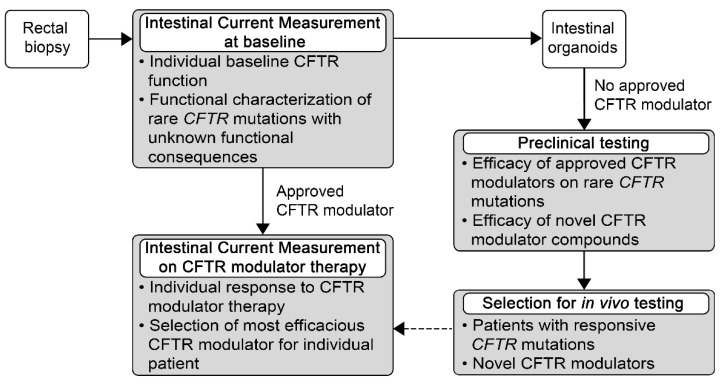
Use of intestinal current measurement for personalized medicine of CF. Intestinal current measurement (ICM) in rectal biopsies can determine individual baseline cystic fibrosis (CF) transmembrane conductance regulator (CFTR) function and provide functional characterization of rare mutations with unknown functional consequences. In CF patients with *CFTR* genotypes that can be treated with approved CFTR modulators, ICM can be used to determine the individual response to therapy at the level of the CF basic defect. If multiple approved CFTR modulator therapies are available, ICM can help to find the most efficacious drug for the individual patient. However, as ICM detects the improvement of in vivo CFTR function, patients needed to be treated with the different CFTR modulator therapies and repeated rectal biopsies would be necessary to compare responses to these different drug regimen. In patients with rare *CFTR* mutations and genotypes with no approved CFTR modulator therapies, rectal biopsies obtained for ICM can be used to generate intestinal organoids for preclinical testing of (i) responsiveness of rare mutations to approved CFTR modulators; and (ii) evaluation of promising novel compounds. ICM may help to verify results from preclinical studies in n-of-1 studies or early phase clinical trials.

**Table 1 jpm-11-00384-t001:** Published studies using intestinal current measurement (ICM) as a biomarker of cystic fibrosis transmembrane conductance regulator (CFTR) function.

Category	Reference
Development of ICM	[15,16,17,18,19,21,22,24,26]
ICM as a diagnostic test for CF	[28,29,30,31,40,42,47,49,50,53,54,75,80]
ICM as outcome measure of preclinical ex vivo response to CFTR-directed therapeutics	[25,43,48]
ICM as outcome measure of in vivo response to CFTR-directed therapeutics	[44,45,46,76]

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
