# Peer review of "Potential of Intestinal Current Measurement for Personalized Treatment of Patients with Cystic Fibrosis"

_jpm, 2021, doi:10.3390/jpm11050384_

Round 1

Reviewer 1 Report

In the review by Graeber et al., the authors examined the use of intestinal current measurement (ICM) as a valuable biomarker to support diagnostic testing and may enhance the prediction of clinical response of CFTR modulators in Cystic Fibrosis patients. Moreover, the rectal tissue collected for ICM could be used as preclinical tool to test current and novel CFTR modulators for rare CFTR mutations. The review is well-written but this reviewer has some comments:

  • The authors mention “CFTR modulators” and cited clinical trials of ORK, SYM and TRIK without any description of them. Please include a paragraph talking about CFTR modulators (correctors, potentiators and amplifiers). Many different labs (Amaral, Bear, Beekman, Galietta, Gentzsch, Lukacs, Rowe) used patient-derived tissues to test FDA-approved and novel CFTR modulators as a preclinical tool in personalized medicine.
  • Preparing a table with the published finding of ICM  as a biomarker of CFTR function may help the readers understand its use better.
  • Line 178: There are preclinical studies supporting that TRIKAFTA rescued rare CFTR mutations in patient-derived tissue. Please include a sentence about this with the following citations (doi: 10.1172/jci.insight.139983; 10.1183/13993003.02774-2020).
  • Lines 181-184: please includes the finding by our colleagues that have been demonstrated functional rescue of rare CFTR mutations with novel CFTR modulators (doi: 10.1038/s41591-018-0200-x, 10.1016/j.jcf.2020.07.015, 10.15252/emmm.201607137, 10.1016/j.jcf.2020.07.003)
  • Line 197: please include the following citations: 10.1002/humu.23741, 10.3390/jpm10020040, 10.1016/j.jcf.2016.09.005, 10.1016/j.jcf.2019.12.001, 10.1016/j.ebiom.2014.12.005, 10.3390/jpm1004020, 10.1016/j.jcf.2020.07.003).

Author Response

Point 1: In the review by Graeber et al., the authors examined the use of intestinal current measurement (ICM) as a valuable biomarker to support diagnostic testing and may enhance the prediction of clinical response of CFTR modulators in Cystic Fibrosis patients. Moreover, the rectal tissue collected for ICM could be used as preclinical tool to test current and novel CFTR modulators for rare CFTR mutations. The review is well-written but this reviewer has some comments:

Response 1: We thank the reviewer for the positive evaluation of our review and have carefully revised our manuscript to address the remaining comments raised by the reviewers in the text and point by point response below.

Point 2: The authors mention “CFTR modulators” and cited clinical trials of ORK, SYM and TRIK without any description of them. Please include a paragraph talking about CFTR modulators (correctors, potentiators and amplifiers). Many different labs (Amaral, Bear, Beekman, Galietta, Gentzsch, Lukacs, Rowe) used patient-derived tissues to test FDA-approved and novel CFTR modulators as a preclinical tool in personalized medicine.

 Response 2: We have now included a paragraph introducing CFTR modulators in more detail and acknowledge the work by our colleagues on FDA-approved and novel CFTR modulators in patient-derived models systems in the preclinical setting for rare mutations (Lines 179 - 189).

 Point 3: Preparing a table with the published finding of ICM as a biomarker of CFTR function may help the readers understand its use better.

Response 3: We thank the reviewer for this suggestion and have now included a table with the published studies using ICM as a biomarker of CFTR function (Table 1).

 Point 4: Line 178: There are preclinical studies supporting that TRIKAFTA rescued rare CFTR mutations in patient-derived tissue. Please include a sentence about this with the following citations (doi: 10.1172/jci.insight.139983; 10.1183/13993003.02774-2020).

Response 4: We have included a sentence about the effect of Trikafta on rare CFTR mutations and cited the mentioned studies (Lines 213 - 216).

 Point 5: Lines 181-184: please includes the finding by our colleagues that have been demonstrated functional rescue of rare CFTR mutations with novel CFTR modulators (doi: 10.1038/s41591-018-0200-x, 10.1016/j.jcf.2020.07.015, 10.15252/emmm.201607137, 10.1016/j.jcf.2020.07.003)

Response 5: We have included the findings about the effect of novel CFTR modulators on rare CFTR mutations and cited the mentioned studies (Lines 213 - 216).

 Point 6: Line 197: please include the following citations: 10.1002/humu.23741, 10.3390/jpm10020040, 10.1016/j.jcf.2016.09.005, 10.1016/j.jcf.2019.12.001, 10.1016/j.ebiom.2014.12.005, 10.3390/jpm1004020, 10.1016/j.jcf.2020.07.003).

Response 6: The studies are now cited as suggested (Line 213).

Reviewer 2 Report

This paper by Graeber et al is a very clear literature review that summarizes a very important topic in the Cystic Fibrosis field. This article highlights the use of intestinal current measurements (ICM) as a technique to quantify CFTR function to be used as a diagnosis tool, as well as to assess CFTR modulators effect. This review shows how ICM can be used as a valuable biomarker in CF, and I strongly recommend it for publication.

 I have just a few minor concerns regarding the manuscript in its present form.

  1. Some additional information has been published by Silva et al 2020 (PMID: 33424627) and might enhance the discussion, since it is the most recent work published using rectal biopsies to perform ICM, and the study using the biggest cohort.
  2. Technical issues and disadvantages of using rectal biopsies in CF diagnosis should be addressed.
  3. Page 4 line 187 the mutations analyzed in [43] should be clarified in this review text.
  4. The fact that ICM has to be performed after a certain drug has to be given to the patient, or several combinations of drugs to determine the best therapeutic option by ICM (which implicates several rectal biopsies procedures) should be addressed or clarified.

Author Response

Point 1: This paper by Graeber et al is a very clear literature review that summarizes a very important topic in the Cystic Fibrosis field. This article highlights the use of intestinal current measurements (ICM) as a technique to quantify CFTR function to be used as a diagnosis tool, as well as to assess CFTR modulators effect. This review shows how ICM can be used as a valuable biomarker in CF, and I strongly recommend it for publication.

Response 1: We thank the reviewer for the positive evaluation of our review and have carefully revised our manuscript to address the remaining comments raised by the reviewers in the text and point by point response below.

 Point 2: Some additional information has been published by Silva et al 2020 (PMID: 33424627) and might enhance the discussion, since it is the most recent work published using rectal biopsies to perform ICM, and the study using the biggest cohort.

Response 2: We thank the reviewer for this suggestion and have included the study to highlight the potential of ICM differentiating between atypical and classical CF (Lines 131-132).

Point 3: Technical issues and disadvantages of using rectal biopsies in CF diagnosis should be addressed.

Response 3: We have included a sentence addressing the limitations on rectal biopsies in CF diagnostics (Lines 137 - 139).

Point 4: Page 4 line 187 the mutations analyzed in [43] should be clarified in this review text.

 Response 4: In this study, patients with residual CFTR function with a broad range of CFTR mutations (Y161C, V232D, R334W, T338I, I1234V, 3272-26 A>G, 3849+10 kb C>T, 4005+5727 A>G, G576A, F1052V, M1137R, 1898+3 A>G) were included. We have now clarified this aspect in the manuscript (Lines 201 - 203).

Point 5: The fact that ICM has to be performed after a certain drug has to be given to the patient, or several combinations of drugs to determine the best therapeutic option by ICM (which implicates several rectal biopsies procedures) should be addressed or clarified.

 Response 5: We have added a sentence to clarify the issue raised by the reviewer (Lines 260 - 262)

Round 2

Reviewer 1 Report

All of my concerns have been addressed